# Metagenomic Insight into the Community Structure of Maize-Rhizosphere Bacteria as Predicted by Different Environmental Factors and Their Functioning within Plant Proximity

**DOI:** 10.3390/microorganisms9071419

**Published:** 2021-06-30

**Authors:** Saheed Adekunle Akinola, Ayansina Segun Ayangbenro, Olubukola Oluranti Babalola

**Affiliations:** Food Security and Safety Niche, Faculty of Natural and Agricultural Sciences, North-West University, Private Mail Bag X2046, Mmabatho 2735, South Africa

**Keywords:** plant-microbe interactions, high-throughput pyrosequencing, soil edaphic factors, nutrient pathways, stress response

## Abstract

The rhizosphere microbiota contributes immensely to nutrient sequestration, productivity and plant growth. Several studies have suggested that environmental factors and high nutrient composition of plant’s rhizosphere influence the structural diversity of proximal microorganisms. To verify this assertion, we compare the functional diversity of bacteria in maize rhizosphere and bulk soils using shotgun metagenomics and assess the influence of measured environmental variables on bacterial diversity. Our study showed that the bacterial community associated with each sampling site was distinct, with high community members shared among the samples. The bacterial community was dominated by Proteobacteria, Actinobacteria, Acidobacteria, Gemmatimonadetes, Bacteroidetes and Verrucomicrobia. In comparison, genera such as *Gemmatimonas*, *Streptomyces*, *Conexibacter*, *Burkholderia*, *Bacillus*, *Gemmata*, *Mesorhizobium*, *Pseudomonas* and *Micromonospora* were significantly (*p* ≤ 0.05) high in the rhizosphere soils compared to bulk soils. Diversity indices showed that the bacterial composition was significantly different across the sites. The forward selection of environmental factors predicted N-NO_3_ (*p* = 0.019) as the most influential factor controlling the variation in the bacterial community structure, while other factors such as pH (*p* = 1.00) and sulfate (*p* = 0.50) contributed insignificantly to the community structure of bacteria. Functional assessment of the sampling sites, considering important pathways viz. nitrogen metabolism, phosphorus metabolism, stress responses, and iron acquisition and metabolism could be represented as Ls > Rs > Rc > Lc. This revealed that functional hits are higher in the rhizosphere soil than their controls. Taken together, inference from this study shows that the sampling sites are hotspots for biotechnologically important microorganisms.

## 1. Introduction

The symbiotic relationship between soil microbiome and plants enables easy adaptation of plants to environmental changes. Within the proximity of plants, microorganisms enhance plant growth by inducing direct and indirect mechanisms, such as plant pathogen suppression, enhanced nutrient availability and tolerance to environmental stressors, such as drought and salinity [1,2]. In return, plant produces compounds such as amino acids, carbon sources, and other root exudates that create an enabling environment for the organisms to thrive [3].

The below-ground environment of plant includes the root endosphere (within the epidermis of the plant tissue), rhizosphere soil (an area influenced by plant), and bulk soil (an area not influenced by plant). The plant endosphere and rhizosphere are usually shaped by root exudates from plant roots, which attract distinct microbial communities. A part of the organisms around the root of plant penetrates the plant via the root hair to inhabit the plant’s endosphere [2,4,5]. The ability of these microbes to penetrate the plant is determined by the innate immunity of the plant. Thus, microbe inhabiting the plant tissues are regulated by various mechanisms controlled by the host plant [6,7]. 

Microorganisms from different domains, such as bacteria, fungi, viruses and archaea have been detected across the rhizosphere, phyllosphere and endospheric parts of plants with more emphasis on both bacteria and fungi groups [8]. These domains possess influential traits, such as the mobilization of nutrients and metabolic processes needed for plant growth, health and development [9,10].

Previously, the advancement in agricultural production depends mainly on the use of chemical and synthetic soil amendments, which have adverse effects on the environment. The hinderance imposed because of extensive use of agrochemicals include; low nutrient uptake, reduced soil quality and environmental hazards viz. greenhouse gas emission and eutrophication [11]. To reduce the menace posed by chemical fertilizers, the use of ecofriendly soil improvement strategy is being considered [12]. Currently, the use of often-neglected microorganisms that naturally colonized plants has been the present paradigm adopted by most agriculturists to exploit beneficial traits possessed by soil organisms [13,14]. The use of ecofriendly option will adequately help feed the increasing world population and avoid threats posed by the use of chemical fertilizers. 

Over the past two decades, there have been growing interest in the use of indigenous rhizosphere organisms, especially stress-tolerating organisms, as bioinoculants for treating soil [3,15]. Bioprospection of the rhizosphere of plants led to the discovery of important plant growth-promoting bacteria, most of which have been tested as positive bioinoculants [15,16,17].

The soil physicochemical properties are strong factors influencing structural diversity and functioning of microorganisms in plant proximity [18]. These factors also function as the essential pointer to assess ecological functions and processes, such as promotion of plant growth, mineralization and easy decomposition of soil organic materials. The relationship between functional and structural diversity of soil microbiome in response to soil pedological parameters has been widely studied and yet poorly understood [19]. Even though the soil pH has been reported as the main driver of soil microbial structure and functions, especially bacterial communities, which thrive better in neutral soils, several other factors viz. nitrate, organic matter and sulfate have also been reported helpful in determining structural composition of plant soil [20,21].

Maize is one of the top three most important and widely grown crops serving as the basis of food security in the poorest regions in the world [22]. In a report published by the International Institute of Tropical Agriculture (IITA) in conjunction with International Maize and Wheat Improvement Center (CIMMYT), about 765 million tons of maize were produced in 2010 from less than 153 hectares across the world [23], most of which are either taken as a staple food or used for the production of alcohol, starch, adhesives, flakes, corn meal, oil, glucose and syrup [24]. Maize serves as a good model in genomics because the inter-relationship between soil fertility and physiological responses is easily compared with other trials. Also, maize has broad economic value and serves as the basis of animal nutrition worldwide. 

Several studies have worked on the bacterial community associated with maize plant (roots, stems, flower and leaves) and the endosphere with fascinating reports on the importance of the rhizosphere and endo-rhizosphere organisms in sustaining agroecosystem [25,26,27]. To ensure the colonization of these important soil organisms, maize roots release a variety of important carbon-containing rhizodeposits, such as exudates, border cells, mucilage and nutrients that makes the rhizosphere of maize selective and nutritive than the surrounding soils. The oligotrophic nature of the proximal maize environment ensures changes in soil microbiota. Nonetheless, microbial biodiversity in the rhizosphere has been reported lower than the bulk soils because carbon availability reduces microbial growth in plant’s proximity [28].

Despite South Africa being classified as a semi-arid region, the country is prominently known as a high maize producing nation, exporting an average of 1 million tons of maize annually [29]. As a result, we speculated that the provinces with the largest production of maize in South Africa could be a hotspot for novel microbial communities and their distribution could be subjective to different environmental factors and soil nutrients. To unveil unculturable organisms and explore the reads beyond domain level, we adopted shotgun metagenomics to assess bacterial communities in maize rhizosphere and bulk soils. This method will give a clear picture of the community structure across the sampling sites compared to 16S amplicon sequencing approach [30]. 

## 2. Materials and Methods

### 2.1. Site Description and Soil Sampling

For this study, we sampled two (2) different farms located at Lichtenburg (25°59′40.4″ S 26°31′44.5″ E) and Randfontein (26°11′52.0″ S 27°33′18.3″ E) at North West and Gauteng provinces of South Africa. After obtaining the permission of farm owners, maize rhizosphere and bulk soils were collected from Lichtenburg (sample-Ls, bulk-Lc) and Randfontein (sample-Rs, bulk-Rc) farms at a diameter of 6-cm around the plant, 2 × 4 m^2^ area and 20-cm depth, while the bulk samples were collected from an area 10-m adjacent to each rhizosphere site [31]. After which, samples were transported to the laboratory in a cooler box filled with ice and stored at −20 °C for subsequent analysis.

### 2.2. Physicochemical Analysis of Maize Rhizosphere and Bulk Soils

After soil sampling, 20 g of each sample was air-dried, mixed and sieved using a 2 mm pore sieve to remove dirt. The pH of samples was measured using a pH-meter after mixing with distilled water at a ratio of 1:2.5 soil. The soil organic matter (OM), organic carbon (Org-C) and total carbon (Total C) were determined using the method in Walkley and Black [32] and Shi et al. [33]. According to the method adopted by Deke et al. [34], exchangeable potassium (K) was measured using a flame photometer, while available phosphorus (P) content was obtained using a spectrophotometer. Soil sulfate was determined using the method by Walker [35]. The nitrate (N-NO_3_) content of samples was determined using LAQUA twin nitrate ion meter (B-742) [36], while ammonium (N-NH_4_) content was measured using a modified manual calorimetric procedure for measuring ammonium in soil and plant Kjeldahl digests as adopted by Baethgen and Alley [37]. 

### 2.3. DNA Extraction and Metagenome Sequencing

The soil DNA was extracted from a 5 g sample using DNeasy Power-Max soil kit (MOBIO Laboratories, Carlsbad, CA, USA) according to the manufacturer’s procedure with little amendment. A shotgun whole-genome sequencing was conducted at the Molecular Laboratory of MR DNA (Shallowater, TX, USA) using Illumina sequencing platform. Between 20–50 ng DNA was used to prepare libraries with the aid of a Nextera DNA flex library kit. After which, the samples were simultaneously fragmented, and adapter sequences added. Using the Qubit^®^ dsDNA HS Assay Kit (Life Technologies, Carlsbad, CA, USA), the concentration of the libraries was measured and the average library size was determined using Agilent 2100-Bioanalyzer. The libraries were pooled and diluted to 0.6 nM and the Illumina NovaSeq system was used to sequence the paired-ends for 300 cycles.

### 2.4. Downstream Analysis of Sequences

The raw metagenome sequences were uploaded to the metagenomics rapid annotation subsystem technology (MG-RAST) server and sequences were subjected to quality filtering that involves the use of SolexaQA to trim low-quality reads and dereplicate the metagenome datasets [38]. Subsequently, DRISEE-duplicate reads inferred sequencing error estimation was used to sieve sequencing error due to the measuring of artificial duplicate reads (ADRs), and reads related to model organisms, such as fly, human and mouse were screened using Bowtie aligner [39]. After quality assessment of the reads, the annotation of sequences against other databases on M5NR-database was performed using the BLAST-like alignment (BLAT-algorithm) [40]. In the SEED subsystems database on MG-RAST, bacterial taxonomic classification (˃90%) was sieved, and other domains were screened-out. The mean abundance values of the replicates were sorted and agglomerated with the unclassified reads related to bacterial domain retained for statistical analysis. The raw sequences are publicly available on NCBI SRA database with project identification number PRJNA645385 and PRJNA645371 for Ls and Rs sites, respectively.

### 2.5. Statistical Analysis

During annotation, MG-RAST server was used to plot the rarefaction curve of the sequences. The physicochemical properties between the sampling sites were compared using a one-way analysis of variance (ANOVA) with Tukey’s HSD pairwise comparison test on GraphPad Prism v5. To assess bacterial diversity within the samples, Pielou evenness and Shannon diversity indices were used, while comparison was calculated using Kruskal-Wallis test. Principal coordinate analysis (PCoA) adopting a Euclidean distance matrix was used to depict bacterial structural diversity (β-diversity) of the sampling sites, while the differences between habitats were calculated using analysis of similarities (ANOSIM) on PAST v3.20. The distribution of bacterial phyla across the sites was shown using principal component analysis (PCA), and the relative abundance was visualized using circus software (http://circos.ca/, accessed on 16 September 2020). Canonical correspondence analysis (CCA) was adopted to explain the effect of environmental variables on bacterial distribution. Meanwhile, the Monte Carlo-999 random permutation test was used for the significant difference in each influential variable on the bacterial community structure. Nonetheless, some functional signatures present in the metagenome study were revealed using SEED-subsystem database of MG-RAST.

## 3. Results

### 3.1. Physicochemical Properties of Maize Rhizosphere and Bulk Soils

The mean values of assessed physicochemical variables showed that K and N-NO_3_ of Ls (K—16.29 mgkg^−1^; N-NO_3_—240.00 mgkg^−1^) and Lc (K—16.24 mgkg^−1^; N-NO_3_—243.00 mgkg^−1^) were significantly higher than Rs (K—8.52 mgkg^−1^; N-NO_3_—167.00 mgkg^−1^) and Rc (K—7.38 mgkg^−1^; N-NO_3_—148.50 mgkg^−1^) of the rhizosphere samples and their controls respectively (*p* ≤ 0.05). Considering factors, such as pH, Org C and P, Randfontein samples Rs (pH–6.76, Org C—1.09 and P—257.14 mgkg^−1^) and their controls Rc (pH—6.73, Org C—0.87 and P—206.54 mgkg^−1^) were substantially higher than Ls (pH—5.62, Org C—0.61 and P–50.98 mgkg^−1^) and their controls Lc (pH—5.87, Org C—0.60 and P—65.86 mgkg^−1^) at *p* ≤ 0.05. A mixed result was deduced from the mean values of N-NH_4_ of the sampling sites and could be represented as Rc ˃ Ls ˃ Rs ˃ Lc (*p* = 0.04). Nevertheless, other environmental variables viz. Total C, OM, and sulfate were insignificantly different across the sampling sites (*p* ˃ 0.05) (Table 1). 

### 3.2. Metagenome Dataset of the Sampling Sites

The raw sequences uploaded on MG-RAST database were Rs—14,928,201, Ls—19,276,118, Rc—1,414,053,905 and Lc—14,988,818. After quality control assessment, the retained mean sequences were Rs—13,823,192, Ls—17,596,177, Rc—13,006,005 and Lc—13, 925,537 for the rhizosphere and bulk samples, respectively. Besides, the average G + C content of the sampling sites was 65 ± 9% for all samples. After QC, predicted protein features were Rs—12,427,664, Ls—15,344,917, Rc—11,695,150 and Lc—12,428,891 while identified protein features were Rs—4,732,504, Ls—5,959,395, Rc—4,507,871 and Lc—4,654,996. The richness of assessed metagenomes was depicted using rarefaction analysis of MG-RAST as shown in Appendix A.

### 3.3. Distribution of Bacterial Community in the Sampling Sites

Using MG-RAST SEED subsystem database, twenty-two (22) conspicuous phyla were sorted from bacterial domain. These include; Proteobacteria, Actinobacteria, Acidobacteria, Gemmatimonadetes, Bacteroidetes, Firmicutes, Verrucomicrobia, Planctomycetes, Chloroflexi, Cyanobacteria, unclassified (derived from bacteria), Nitrospirae, Deinococcus-Thermus, Chlorobi, Spirochaetes, Aquificae, Synergistetes, Thermotogae, Lentisphaerae, Candidatus Poribacteria, Chlamydiae and Fusobacteria (Figure 1). Phyla such as Actinobacteria (*p* = 0.05), Proteobacteria (*p* = 0.05), Acidobacteria (*p* = 0.03), Gemmatimonadetes (*p* = 0.01), Synergistetes (*p* ˂ 0.00), Thermotogae (*p* = 0.04), Candidatus Poribacteria (*p* = 0.01) and Fusobacteria (*p* = 0.05) were significantly different across the sampling sites (Appendix A). PCA was used to depict the distribution of identified bacterial phyla between the samples and could be represented as Lc ˃ Rc ˃ Ls ˃ Rs with Randfontein bulk (control) sample having the highest bacterial distribution (Figure 2).

### 3.4. Structural Diversity of Rhizosphere Bacterial Communities across the Sampling Sites

The metagenome study of maize rhizosphere bacteria showed that nine (9) major phyla were above the relative abundance of 1%, while others were ˂1%. These include; Proteobacteria (Ls—42.53%) and Actinobacteria (Rs—41.61%) that were significantly high in the rhizosphere samples while Acidobacteria (Lc—6.53%) and Gemmatimonadetes (Lc—5.35%) were significantly higher in the bulk samples compared to samples collected from maize rhizosphere (*p* ≤ 0.05). Other phyla from this group (˃1%) were insignificantly different across the sampling sites. Considering bacterial phyla of relative abundance ˂1, thirteen (13) phyla were extrapolated from the metagenome study of our samples. Synergistetes (Rs—1.05%), Thermotogae (Rs—0.06%) and Candidatus Poribacteria (Ls—0.04%) were substantially higher in the rhizosphere samples compared to their controls while Fusobacteria (Lc—0.04%) was the most significantly high phylum in the bulk sample compared to the rhizosphere samples (*p* < 0.05) as revealed in Appendix A.

At the class level, twenty-six (26) conspicuous bacterial classes were identified from our metadata (Figure 3a). The major bacterial classes > 1% relative abundance was eleven (11). Actinobacteria (Rs—48.08%), Betaproteobacteria (Ls—11.51%), Gammaproteobacteria (Ls—3.82%), Planctomycetacia (Rs—2.63%) and Acidobacteria (Ls—1.07%) were significantly higher in the rhizosphere samples while Gemmatimonadetes (Lc—6.32%) and Solibacteres (Lc—4.01%) were substantially higher in the bulk samples (*p* ≤ 0.05). At a relative abundance < 1%, Cytophagia (Rc—0.87%) and Ktedonobacteria (Lc—0.49%) were the significantly high bacterial classes belonging to the control samples (*p* ≤ 0.05). Other bacterial classes viz. Alphaproteobacteria, Deltaproteobacteria, Sphingobacteria, Bacilli etc. were insignificantly high across the sampling sites (Appendix A). Furthermore, at the order level, *Actinomycetales* (Rs—46.71%), *Rhizobiales* (Ls—10.13%), *Burkholderiales* (Ls—10.49%), *Gemmatimonadales* (Ls—7.37%), *Sphingomonadales* (Ls—5.04%), *Myxococcales* (Rs—4.18%) and *Pseudomonadales* (Ls—1.18%) were the significantly dominant bacterial group in the rhizosphere samples while *Solibacterales* (Lc—4.68%), unclassified (derived from *Acidobacteria*) (Lc—1.46%), *Acidobacteriales* (Lc—1.25%), *Rhodobacterales* (Rc—1.22%), *Xanthomonadales* (Rc—1.21%) and *Ktedonobacterales* (Lc—0.567%) were significantly higher in the control samples compared to the rhizosphere of maize (Figure 3b and Appendix A).

At the family level, Gemmatimonadaceae (12.08%), Burkholderiaceae (9.68%), Solibacteraceae (7.70%), Bradyrhizobiaceae (8.48%), Sphingomonadaceae (5.10%) and Acidobacteriaceae (2.05%) were the most abundant in Ls, Streptomycetaceae (11.35%), Conexibacteriaceae (9.80%), Planctomycetaceae (5.18%), Micromonosporaceae (4.53%), Rhizobiaceae (3.34%), Pseudonocardiaceae (1.88%) and Chloroflexaceae (1.24%) were most abundant in Rs, while Nocardioidaceae, Mycobacteriaceae, unclassified Acidobacteria, Frankiaceae, Myxococcaceae, Micromonosporaceae, Geodermatophilaceae, Caulobacteraceae, Polyangiaceae, Xanthomonaclaceae, Micrococcaceae, Methylobacteriaceae, Verrucomicrobia subdivision 3, unclassified (Sphingobacteriales) etc. were most abundant in the control samples (Figure 3c). A high significant difference was observed in family such as Gemmatimonadaceae (*p* = 0.01), Solibacteraceae (*p* = 0.05), Streptomycetaceae (*p* < 0.01), Sphingomonadaceae (*p* = 0.01), Nocardioidaceae (*p* = 0.03), Planctomycetaceae (*p* < 0.01) etc. (Appendix A). 

At the genus level, genera such as *Gemmatimonas* (Ls—16.03%), *Streptomyces* (Rs—14.78%), *Conexibacter* (Rs—13.59%), *Burkholderia* (Ls—10.50%), *Gemmata* (Rs—2.44%), *Micromonospora* (Rs—2.95%), *Pseudomonas* (Ls—2.23%), *Bacillus* (Rs—1.49%), *Streptosporangium* (Rs—1.49%) and *Mesorhizobium* (Rs—1.03%) were significantly higher in rhizosphere samples compared to their controls while genera viz. *Candidatus Solibacter* (Lc—10.22%), *Nocardioides* (Rc—7.23%), *Arthrobacter* (Rc—3.61%), *Sorangium* (Rc—3.06%), *Candidatus Koribacter* (Lc—2.85%), *Methylobacterium* (Rc—2.48%), *Chitinophaga* (Rc—2.37%), *Chthoniobacter* (Rc—1.93%), *Acidobacterium* (Lc—1.77%), *Rhodococcus* (Rc—1.43%) and *Anaeromyxobacter* (Rc—1.31%) were significantly high in the control samples (*p* ≤ 0.05). Others such as *Mycobacterium*, *Bradyrhizobium*, *Frankia*, *Variovorax*, *Kribbella*, *Rubrobacter*, *Sphingomonas*, *Amycolatopsis*, *Sphingobium*, *Nitrospira* and *Catenulispora* were insignificantly different across the sampling sites (*p* ˃ 0.05) (Figure 3d and Appendix A).

### 3.5. Diversity Indices of Bacterial Communities across the Sampling Sites

Pielou evenness and Shannon diversity were used to assess bacterial community variation of rhizosphere samples and their controls. Alpha diversity of bacterial communities showed insignificant differences between taxonomical cadre (*p* > 0.05) using Kruskal-Wallis one-way analysis of variance (Table 2). Beta diversity of bacterial community (phylum) was depicted using Bray-Curtis dissimilarity PCoA and confirmed using ANOSIM. Principal coordinate analysis (PCoA) revealed that the rhizosphere bacteria were widely separated except Rs and Rc samples that were close to each other (Figure 4). The inference from PCoA was confirmed using ANOSIM (*p* = 0.01, *R* = 0.58). Figure 4 revealed that the bacterial community structure between the sampling sites was not weak. Apart from Randfontein rhizosphere soils (Rs 1–3) and the bulk samples (Rc 1–3) that were closely related, Lichtenburg rhizosphere soils (Ls 1–3) and bulk samples (Rc 1–3) were widely separated as confirmed by analysis of similarity (Figure 4).

### 3.6. Influence of Environmental Variables on Bacterial Community

The interaction between assessed physicochemical variables and the relative abundances of bacterial phylum was depicted using Canonical correspondence analysis (CCA). Three (3) parameters, such as sulfate, pH and N-NO_3_ were selected based on their contribution to bacterial distribution and significant test using forward selection of environmental variables (Figure 5). 

The results from CCA revealed that the bacterial community structure of our sampling sites is highly influenced by soil physicochemical factors with CCA permutation test = 0.00. Fusobacteria, Candidatus Poribacteria, Acidobacteria, Proteobacteria, Firmicutes, Aquificae, Nitrospirae and Gemmatimonadetes positively correlated with N-NO_3_ and negatively with sulfate and pH. Likewise, Chloroflexi, unclassified derived from bacteria, Spirochaetes, Chlorobi, Planctomycetes, Cyanobacteria, Synergistetes, Deinococcus-Thermus, Verrucomicrobia, Lentisphaerae and Thermotogae positively correlated with pH and negatively with N-NO_3_ and sulfate. Additionally, Chlamydiae, Bacteroidetes and Actinobacteria positively correlated with sulfate but against N-NO_3_ and pH (Figure 5). The Monte-Carlo permutation test with 999 random sorting and forward selection of environmental variables was used to test the factors that best explain the variation in the bacterial community. CCA showed that N-NO_3_ significantly (*p* = 0.019) contributed 82.70% of the variation, sulfate insignificantly (*p* = 0.50) contributed 9.70% variation, while pH insignificantly (*p* = 1.00) contributed 7.60% of the variation of the bacterial community in the sampling sites (Table 3).

### 3.7. Metagenome-Based Functional Signatures of Bacteria Associated with the Maize Rhizosphere and Bulk Soils

Using SEED subsystem levels 2 and 3, the metagenome study of our samples addressed habitat specificity of bacterial functions in the maize rhizosphere and bulk soils. To do so, functional dispersal of the bacterial metagenomes of each sampling sites were compared. Generally, 31% of the assigned functions belong to Ls, followed by Rs (29%), Lc (22%) and Rc (18%). In this study, we addressed functional signatures such as nitrogen metabolism (NM), phosphorus metabolism (PM), iron acquisition and metabolism (IAM) and stress response (SR).

The NM pathway entails functional hits such as Ls—88,948, Rs—74,740, Lc—69,607 and Rc—73,260 for each sampling sites. The relative abundance of subdivisions deduced from NM viz. Allantoin utilization (Ls—3.43), ammonia assimilation (Ls—25.60), glutamate and aspartate uptake in bacteria (Ls—4.88), poly-gamma-glutamate biosynthesis (Ls—2.65), nitrate and nitrite ammonification (Rs—20.98), nitric oxide synthase (Rs—6.96) and nitrogen fixation (Rs—2.35) were significantly high in rhizosphere samples compared to their controls while Glutamine synthetases (Rc—2.50), denitrification (Lc—5.40) and nitrosative stress (Rc—1.93) were substantially high in the control samples (*p* ≤ 0.05).

The functional hits mined from PM include; Ls—55,847, Rs—44,263, Lc—44,281 and Rc—43,346. Sorted PM subdivisions such as Alkylphosphonate utilization (Ls—6.48), P-uptake by cyanobacteria (Ls—8.07), phosphate metabolism (Rs—62.29) and phosphonate metabolism (Ls—1.25) were significantly high in the rhizosphere samples while other PM related pathways such as PhoR-PhoB component regulatory system, phosphoenolpyruvate phosphomutases, alkylphosphonate (TC_3.A.1.9.1) and high affinity phosphate transporter and control of PHO regulon are insignificantly different across the sampling sites (*p* > 0.05). 

The functional hits mined from IAM include; Ls—26,796, Rs—21,718, Lc—18,688 and Rc—20,450. Classification of IAM viz. siderophore_Aerobactin (Rs—0.18), siderophore_Assembly kit (Rs—8.21), Vibrioferrin synthesis (Ls—0.17), ABC type iron transport system (Ls—3.26), iron scavenging cluster in *Thermus* (Rs—1.26), ABC transporter (Rs—0.98) and iron acquisition in *Vibrio* (Ls—31.89) were significantly high in rhizosphere samples while siderophore_staphylobactin (Lc—0.07), transport of iron (Rc—12.30), *Campylobacter* iron metabolism (Lc—13.70), Gram-negative hemen hemin uptake and utilization (Lc—9.00) and iron acquisition in *Streptococcus* (Rc—2.68) were significantly high in the control samples (*p* ≤ 0.05). Other functions such as siderophore_achromobactin, siderophore_desferrioxamine_E, siderophore_enterobactin, siderophore_pyoverdine, siderophore_staphylobactin, siderophore_yersiniabactin_biosynthesis were insignificantly different across the sampling sites (Table 4).

Furthermore, SR functional signatures were also mined with hits such as Ls—86,683, Rs—68,762, Lc—66,309 and Rc—66,096 deduced from the metagenome study of our sampling sites. Stress responses, such as choline and betaine uptake and biosynthesis (Rs—11.03), ectoine synthesis and regulation (Ls—1.23), phage shock protein operon (Ls—1.54), CoA-desulfide reductases (EC_1.8.1.14) (Rs—1.33), glutathione non-redox reaction (Ls—4.76), formaldehyde detoxification (Ls—1.91), tellurite resistance chromosomal determinants (Ls—0.03) and cold shock *CspA* family of proteins (Ls—3.11) were substantially high in the rhizosphere samples (*p* ≤ 0.05) while osmoregulation (Rc—2.08) and glutathione biosynthesis (Lc—9.82) were significantly high in the control samples (Table 4).

## 4. Discussion

Soil microbiome perform complex activities with important agricultural and ecological benefits. The most important of these functions is the biogeochemical cycling of nutrients and the sustenance of healthy soil via indirect plant-beneficial mechanisms [41]. The soil microbial communities and their functions viz. microbial adsorption, production of extracellular compounds, assisted-proliferation of other important organisms and availability of nutrients are influenced by soil edaphic factors [33]. In this study, we evaluated the influence of soil pedological factors on the diversity of rhizosphere bacterial communities and their functions within maize proximity using shotgun metagenomics. 

Most environmental studies on the specific microbial group adopt PCR-dependent metagenomics to assess the microbial population within the samples. This technique involves the use of either internal transcribed spacer (ITS) or 16S rRNA-dependent metagenomic sequencing approach [42,43]. However, the use of PCR-dependent metagenomic analysis has been reported as biased owing to its unequal amplification of sequences, while a shotgun metagenomics could easily annul this constraint and unveil rare organisms [44]. The metagenome study of our sampling sites showed approximately 89% rhizobacteria using subsystem analysis of MG-RAST. 

This study showed twenty-two (22) major phyla, as deduced from the metadata. Proteobacteria, Actinobacteria, Synergistetes, Thermotogae and Candidatus Poribacteria were significantly high in the rhizosphere samples (Rs and Ls), while Acidobacteria, Gemmatimonadetes and Fusobacteria were significantly high in the bulk samples. The outcome of this research is similar to the studies by Yousuf et al. [45]; Chen et al. [46] and Chica et al. [47] on *Arachis hypogaea*, maize, soybean and tuber crops, respectively. 

At the genus level, Gemmatimonas, Burkholderia, Streptomyces, Conexibacter, Gemmata, Micromonospora, Pseudomonas, Bacillus, Streptosporangium and Mesorhizobium were conspicuously high in the rhizosphere samples while genera such as Candidatus Solibacter, Candidatus Koribacter, Arthrobacter, Nocardioides, Methylobacterium, Chitinophaga, Acidobacterium, Rhodococcus and Anaeromyxobacter were substantially high in the bulk samples. Some of these genera have been reported in studies of Walters et al. [26], Berlanas et al. [48] and Chica et al. [47] on maize rhizobiome. Oftentimes, most of the aforementioned genera associated with the rhizosphere samples have been involved in the biogeochemical cycling of nutrients. An example is Burkholderia gladioli MEL01 that was found to be effective as a chitosanolytic phosphate-solubilizer after fermentation with chitosan [49]. Also, the nutrient solubilizing effect of species from other genera viz. Bacillus, Streptomyces, Pseudomonas and Mesorhizobium have been reported in several studies on maize, chicken pea etc. [17,50,51,52].

In addition, the rhizoremediation effects of *Pseudomonas* and *Bacillus* species have also been reported on diesel-polluted soil and the discovery of redundancy genes encoding enzymes, such as *AlkB*, *LadA* and *CYP450* that possess ring-cleaving dioxygenase and different hydroxylating enzymes involved in the degradation of polyaromatic hydrocarbons were also discovered [53]. The high relative abundance of *Bacillus*, *Pseudomonas* and *Methylobacterium* in Rs and Rc might be attributed to the proximity of Randfontein samples to mining area, which would have accumulated high concentrations of heavy metals emanating from the mine pit. These microorganisms inhabit mine areas [53] and were involved in the biosorption of heavy metals [54]. Nonetheless, our study also revealed unclassified organisms derived from *Acidobacteria*, *Spartobacteria* and *Sphingobacteria* that could highlight the complexity of the samples.

The effect and abundance of bacterial community in the sampling sites was depicted using PCA. Principal component analysis showed that each sample is predominated by different bacterial groups with a combined variation of 89.75% between maize rhizosphere and bulk samples. The vector arrows showed the bacterial phylum that strongly influences the distribution (Figure 2). For instance, Proteobacteria and Fusobacteria predominate Ls (1–3), Actinobacteria and Chlamydiae predominates Rs (1–3) etc. The vector length of bacterial dominance could interfere with the activities and functioning across the site. The abundance of Actinobacteria, Proteobacteria and Fusobacteria in the rhizosphere samples showed high contribution to nutrient cycling, as confirmed by the report of Ghazouani et al. [55]; Malisorn et al. [56] and Seballos et al. [57]. Meanwhile, the bulk samples also harbor important bacterial groups such as Gemmatimonadetes, Acidobacteria, Deinococcus-Thermus, Verrucomicrobia, and Cyanobacteria, which have been reported to be immensely involved in both direct and indirect plant growth-promoting mechanisms [57]. The high population of microorganisms in the bulk samples is expected because it has been reported that there is always a reduced microbial diversity at the rhizosphere soil of plants compared to bulk soil, but with more activity [58,59].

Pielou evenness and Shannon diversity indices were insignificantly different (*p* ˃ 0.05) at different bacterial levels when comparing within samples (Table 2). The PCoA revealed that the structural diversity of bacterial communities across the sampling sites (β-diversity) was significantly different except with Rs (1–3) and Rc (1–3) that were close, as shown in Figure 4. The separations between samples were confirmed using ANOSIM (*p* = 0.01 and *R* = 0.58). The wide difference in the physicochemical properties could constitute to the variations (Table 1). The diversity indices correlate with the findings of Praeg et al. [60] except for the closeness of Rs and Rc samples.

The canonical correspondence analysis (CCA) showed the relationship between the relative abundance of bacterial communities and assessed environmental variables (Figure 5). The vector length of the physicochemical parameters (N-NO_3_, sulfate, and pH) revealed that the parameters considerably influence the community structure of bacteria across the sampling sites. The N-NO_3_ of samples significantly (*p* = 0.019) contributed 82.70% of the variation, sulfate insignificantly (*p* = 0.50) contributed 9.70% variation while pH insignificantly (*p* = 1.00) contributed 7.60% variation in the bacterial community (Table 3). As illustrated by CCA, N-NO_3_ remains the most influential soil factor shaping the structural diversity of bacteria in our sampling sites. Although the significance of pH on bacterial structure has been reported in several studies [61,62], because the pH of our sampling sites was within the considerable range required for bacterial inhabitation, it led to its negligible contribution to the community structure of bacteria of the sampling sites [18]. 

The inference deduced from CCA analysis of the sampling sites showed distinct distribution of bacterial phyla predicted using selected environmental variables. N-NO_3_ (*p* = 0.019) being the most important pedological factor significantly contributed 82.70% variation in the bacterial community considering the vector length (Figure 5). The metagenomic study of the sampling sites also revealed important bacteria viz. *Streptomyces*, *Nitrospira*, *Conexibacter* and *Mycobacterium* (Figure 3d and Appendix A). Strains from these genera are well-known plant growth-beneficial bacteria with distinct traits useful in bioremediation, stress reduction, soil nutrient availability and suppression of soil-borne phytopathogens [3].

The functional studies on maize rhizosphere and bulk soil addressed bacterial functions such as nitrogen metabolism, phosphorus metabolism, stress responses and iron acquisition and metabolism. Each site harbors distinct functional signatures with hits mined from the study represented as Ls ˃ Rs ˃ Rc ˃ Lc. The inference from this study echoes the research of Molefe et al. [63] on the bacterial community structure and functions at the rhizosphere of maize plant, but the outcome differs from the findings of Taffner et al. [43] that compared results from both shotgun and 16S rRNA-metagenomic analysis of archaeal functions associated with soil, rhizosphere and phyllosphere habitats of *Eruca sativa Mill*.

Nitrogen is an important macronutrient for the sustainability of life, and plants are not exempt. Nitrogen gas is readily available in the atmosphere and biologically needed. Plants acquire N_2_ in the soil for metabolic activities in the form of fixed molecules viz. nitrate, amino acids ammonia and urea from nitrogen-fixing bacteria. Adequate N_2_ in the soil enhances an increase in plant biomass and other physiological attributes [64]. Our metagenome study showed fourteen (14) major pathways involved in nitrogen metabolism. Allantoin utilization, ammonia assimilation, glutamate and aspartate uptake in bacteria, poly-γ-glutamate biosynthesis, nitrate and nitrite ammonification, nitric oxide synthase production and nitrogen fixation were substantially high in the rhizosphere samples (Ls and Rs) while glutamine synthases, denitrification and nitrosative stress were abundant in the bulk soils (*p* ≤ 0.05). 

Allantoin are purine-derived ureides with numerous nitrogenous materials mainly transported to the aerial of plant via root nodules. Allantoin utilization ensures nitrogen recycling for plant growth. Allantoin utilization is well represented in our metagenome study and has been reported to play a major role in nutrient cycling and stress tolerance [65]. The use of allantoin in relation to the formation of nodules in soybean was reported by Tajima et al. [66]. The study confirmed the importance of allantoin for seed formation and efficient growth of soybean. Likewise, the importance of *PvUPS1* allantoin transporter in the nodulation of soybean was also studied by Pélissier et al. [67] with the strongest expression of *PvUPS1* noticed in the root of nodulated plants compared to the non-nodulated roots. 

Active components of ammonia assimilation mined from our studies include; glutamate, aspartate, glutamine and asparagine biosynthesis, glutamate hydrogenase and poly-γ-glutamate biosynthesis. These pathways involve the fixing of ammonia into organic compounds and enhanced formation of amino acids and amides from NH_3_ and keto acids. The complex degradation involves two routes; the reduction of amine of a keto acid to produce amino acid in the presence of glutamate dehydrogenase (E.C.1.4.1.3), the incorporation of NH_3_ into glutamine by glutamine synthetase (E.C.6.3.1.2) and transfer of the amide-amino group into α-ketoglutarate by α-ketoglutarate amino transferase (E.C.2.6.1.53) that is readily available for plant [68,69]. 

Other well-represented pathways involved in the process of nitrogen metabolism include nitrate and nitrate ammonification, nitric oxide (NO) synthase, production of nitrilase and nitrogen fixation. These pathways are most abundant in the rhizosphere samples and correlates with the findings of Mendes et al. [70] that compared nitrogen metabolism pathways involved in rhizosphere and bulk soils of soybean. The abundance of large contingents involved in nitrogen fixation are logical properties of most plant environments [70]. However, nitrosative stress function was also mined from the sampling sites but most abundant in the bulk soils. Nitrosative stress induces overproduction of NO and NO-related products, which could be toxic to the physiological processes of plants [71]. 

Phosphorus (P) is another important plant macronutrient. P is a key biomolecule involved in energy metabolism viz. ATP, pyrophosphate, phospholipids and nucleic acids. P is oftentimes limited in the soil and can be easily absorbed by plants, promoting the widely use of phosphorus fertilizer for the augmentation of plant soil [72]. Nine (9) major pathways related to phosphorus metabolism were mined from our sampling sites. These include; alkylphosphonate utilization, P uptake by cyanobacteria, alkylphosphonate (TC_3.A.1.9.1), phosphate and phosphonate metabolism that were conspicuously abundant in the rhizosphere samples, while other pathways were insignificantly different between the sampling sites. Mendes et al. [70] reported a similar result on the abundance of alkylphosphonate and P-uptake in the rhizosphere soil of soybean. Other functions detected from the sampling sites, as mentioned above, indicated an enhanced plant P-availabilty through mineralization, P-solubilization and reduced soil pH [73]. 

Iron is an important nutrient involved in physiological processes such as respiration, photosynthesis, gene regulation and oxygen transport in plants and soil microorganisms [74]. This study revealed twelve (12) conspicuous iron acquisition and metabolism pathways, and eleven (11) other siderophore-related rhizosphere bacterial functions involved in maize rhizosphere and bulk soils. Considering the siderophore-related functions, sideraphore_aerobactin, vibrioferrin_synthesis and siderophore assembly kits were substantially high in the rhizosphere samples while siderophore_enteribactin is most abundant in the bulk samples (*p* ≤ 0.05). In the case of other pathways, iron scavenging in *Thermus*, ABC type iron transport, B12 siderophore hemin and iron acquisition in *Vibrio* were significantly high in the rhizosphere samples while iron transport, *Campylobacter* iron metabolism, hemen hemin-uptake and iron acquisition in *Streptococcus* were majorly involved in the bulk soils (*p* ≤ 0.05). The aforementioned processes have been confirmed useful in Fe-uptake by plants. The result also inferred an intense rivalry for Fe-uptake [73]. According to correlation statistics performed by Mendes et al. [70], bacterial phyla associated with the above highlighted functional traits are Proteobacteria, Actinobacteria, Verrucomicrobia and Planctomycetes. Meanwhile, most of the above-mentioned phyla are relatively abundant in the rhizosphere samples (Ls and Rs) (Figure 1 and Appendix A). 

Plant stress responses narrate the complex cellular and molecular processes induced by a result of shift in the normal attributes of soil edaphic factors. Plant stress factors can either be biotic viz. pathogens and herbivores or abiotic, such as osmotic, oxidative, drought stressors etc. These abnormalities tend to reduce the productivity of plants by activating harsh crop impairment factors [3]. Eight (8) major pathways related to soil stress reduction and several subunits associated with osmotic, oxidative, acid and toxin-related responses were deduced from the sampling sites. Choline and betaine uptake and biosynthesis, ectoine synthesis and regulation, phage shock protein operon, CoA disulfide reductase (EC_1.8.1.14) cluster, glutathione biosynthesis, formaldehyde detoxification pathway, tellurite resistance determinants and cold shock *CspA* family proteins were significantly more abundant in the rhizosphere samples while osmoregulation and glutathione-related pathways were significantly high in the bulk samples (*p* ≤ 0.05). These pathways have been reported viable against plant’s soil stressors by studies viz. Ali et al. [75]; Annunziata et al. [76] and Tang et al. [77]. The effect of choline and glycine betaine in osmoregulation of important rhizobiome strains useful in salt tolerance was explored by Boncompagni et al. [78]. Also, the potential of *Pseudomonas protegens* SN15-2 in osmotic stress resistance as a result of *betA* and *betB* (involved in glycine-choline betaine pathway) production was investigated by Tang et al. [77], and thus suggested the field application of *P. protegens* SN15-2 in crop improvement. Rubrerythrin and glutaredoxins stress reducing pathways were also well represented in this study. Rubrerythrin oxidative stress regulation involves the use of *RbrA* enzyme to decompose H_2_O_2_. In a research reported by Zhao et al. [79], *RbrA* enzyme was described useful in the protection of nitrogenase damage induced by H_2_O_2_ in *Anabaena* sp. PCC7120. Glutaredoxins have been discovered useful in three (3) different forms with specific functions. They control protein glutathionylation as an oxidoreductase (apoforms). They help control iron homeostasis by acting as a sensor to cellular iron distribution (holoforms) and regulate the intracellular sequestration and uptake of iron when deficient (isoforms) [80]. 

Selenium is an essential micronutrient to humans and animals. Se deficiency in dietary intake can impose unwanted health disorders and diseases such as reduced immunological functioning and fertility [81,82]. The uptake of selenate and selenite has been well represented in this study and could increase the nutritive value of plants and likewise the detoxification of unwanted toxins [81]. A study conducted by Di Gregorio et al. [83] on *Brassica juncea* showed that soil rhizosphere bacteria augment the uptake and decomposition of selenite oxyanions. However, excessive absorption of Se could inhibit the growth of plants [84]. This research also accounted for the presence of cold shock *CspA* family of proteins and heat shock *dnaK* genes cluster in our metagenome study. These genes have been reported as essential proteins produced by plant growth-promoting bacteria against several plant stressors [85]. A study by Shafqat et al. [86] confirmed the effects of heat shock protein (*dnaK*) and aquaporin expression in the enhancement of citrus growth in high temperature and water deficient conditions, while *CspA* family proteins produced by soil microbiome were also reported crucial in curbing harsh conditions posed by cold shock on plants [87,88]. 

Other important pathways include formaldehyde detoxification and tellurite resistance determinants. Exposure and ingestion of formaldehyde have been reported to be harmful to humans and plants by destroying important biomolecules viz. DNA and protein [89] while tellurite (Te) and its compound were also reported destructive to the growth of plants. The study by Martin [90] showed that tellurium induced a yellow-white mottling chlorosis and reduced yield on wheat grown under low concentration of Te, while increased concentration of Te leads to a reduction in the size of secondary roots and dryness of the plant.

## 5. Conclusions

Adopting a shotgun sequencing approach, this study revealed that Proteobacteria, Actinobacteria, Acidobacteria, Gemmatimonadetes, Bacteroidetes and Verrucomicrobia have substantially predominated in the maize rhizosphere and bulk soils. Additionally, at the genus level, numerous plant-beneficial bacteria were identified in the metagenome study with *Gemmatimonas*, *Streptomyces*, *Conexibacter*, *Bacillus*, *Burkholderia*, *Gemmata*, *Mesorhizobium*, *Pseudomonas* and *Micromonospora* found most abundant in the rhizosphere soils. The community differences across the sampling sites showed that the rhizosphere-bacteria were significantly different across the habitats except Rs and Rc samples that were closely related. A close relationship between soil edaphic factors and bacterial diversity was also noticed with N-NO_3_ observed as the major influential factor to the changes in bacterial communities, while sulfate and pH contributed insignificantly to the community structure of bacteria. The forward selection of environmental variables showed that changes in soil edaphic factors could influence bacterial diversity at plant’s vicinity. Functional signatures considering pathways such as nitrogen metabolism, phosphorus metabolism, stress responses and iron acquisition and metabolism revealed that maize rhizosphere soils have high functional hits than bulk soils. Understanding the mechanisms involved in rhizosphere-bacterial community selection, especially the importance of rhizodeposits in modification of soil microbiome is needed to explore the benefits of agroecosystem.

## Figures and Tables

**Figure 1 microorganisms-09-01419-f001:**
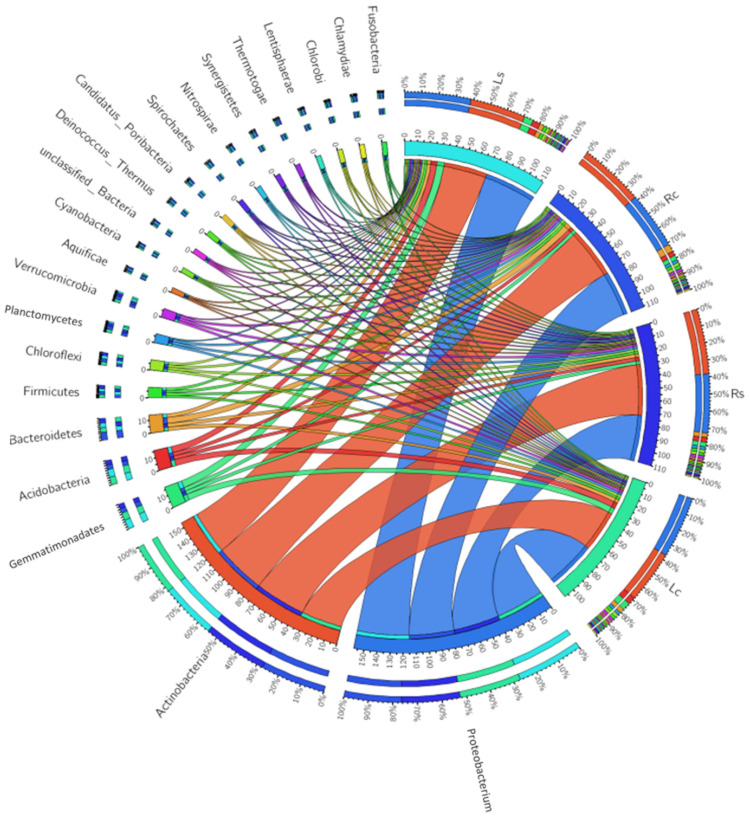
The relative abundance of bacterial phyla in maize rhizosphere and the bulk soils. Ls—Lichtenburg rhizosphere sample, Lc—Lichtenburg bulk (control) sample, Rs—Randfontein rhizosphere sample, Rc—Randfontein bulk (control) sample. In the circus plot, the maize rhizosphere and bulk samples were arranged radially and the percentages of each phylum represented with color chords linking them together. For example, the color chords of Proteobacteria and Actinobacteria were larger than other phyla and their percentages can be viewed at the edge of each sampling sites and controls.

**Figure 2 microorganisms-09-01419-f002:**
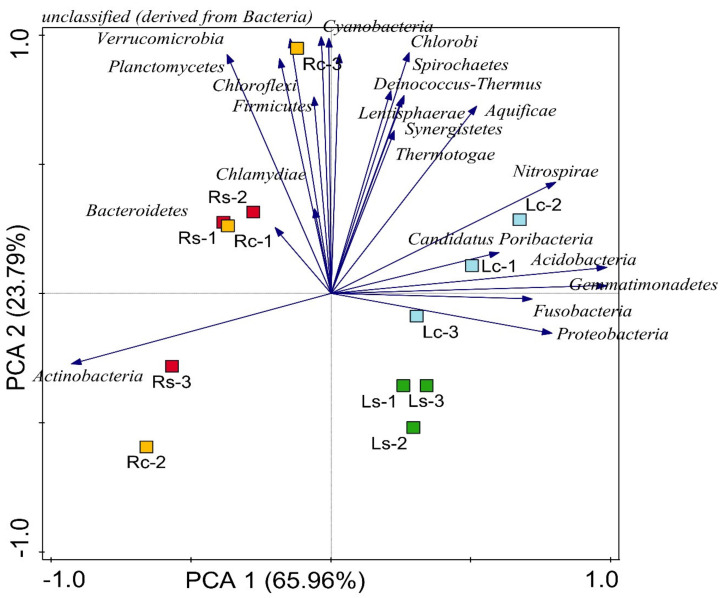
Principal component analysis (PCA) of bacterial metagenomes. The length of each vector represents the influence of the bacterial group metagenomes on the environment. Ls-1–3 (replicates of rhizosphere soil samples collected from Lichtenburg site), Lc-1–3 (replicates of bulk soil samples collected from Lichtenburg site), Rs-1–3 (replicates of rhizosphere soil samples collected from Randfontein site), Rc-1–3 (replicates of bulk samples collected from Randfontein site). The length of vectors shows the strength of influence of each phylum in the soil (e.g., Verrucomicrobia, Planctomycetes, Cyanobacteria and unclassified bacteria had most influence on Rs-1, Rs-2, Rc-1 and Rc-3; Chlorobi, Nitrospirae, Cyanobacteria, Acidobacteria and Gemmatimonadetes had most influence on Lc-1–3; Proteobacteria, Gemmatimonadetes and Acidobacteria had most influence on Ls-1–3 while Actinobacteria had most influence on Rc-2 and Rs-3). Coordinate axis 1 (65.96%) and 2 (23.79%) explain variation in the sampling sites.

**Figure 3 microorganisms-09-01419-f003:**
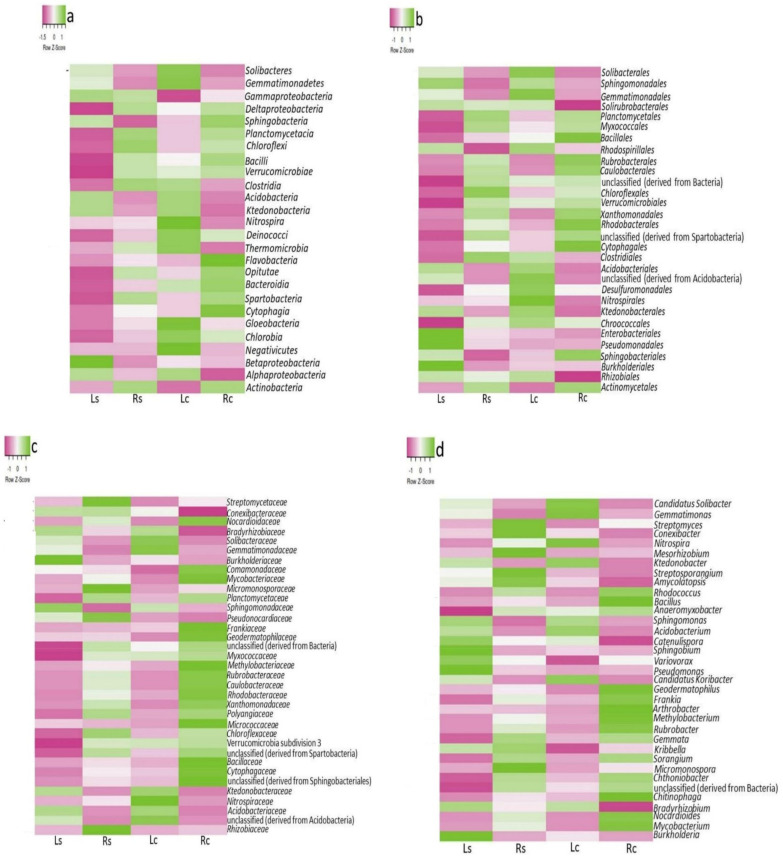
Heatmap of maize rhizosphere bacterial community: Class (**a**), Order (**b**) Family (**c**) and Genus (**d**). The color saturation (z-score) represents the relative abundance of three replicates deduced from the maize rhizosphere (Ls and Rs) and bulk (Lc and Rc) soils. All statistical analyses, including mean values and analysis of variance (ANOVA) were done using GraphPad Prism (v5.0) as shown in Appendix A.

**Figure 4 microorganisms-09-01419-f004:**
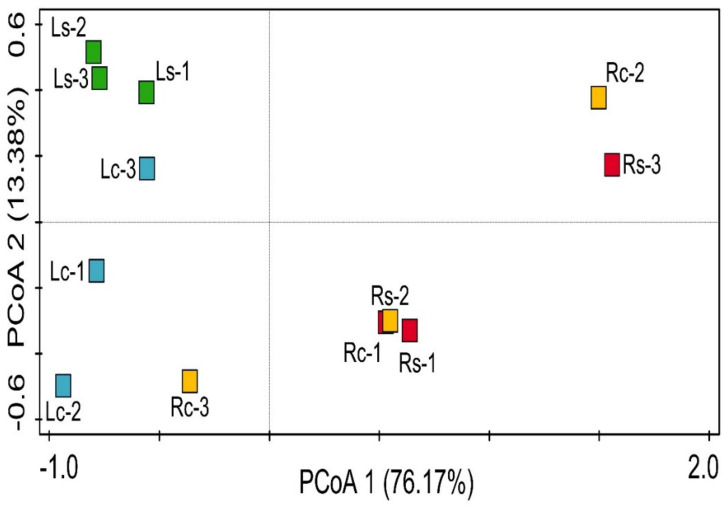
Principal coordinate analysis (PCoA) plot of rhizosphere bacterial communities across sampling sites using Bray-Curtis dissimilarities. Coordinate axis 1 (76.17%) and 2 (13.38%) explain separation in the sampling sites. Ls-1–3 triplicate rhizosphere soil from Lichtenburg site, Lc-1–3 triplicate bulk (control) soil from Lichtenburg site, Rs-1–3 triplicate rhizosphere soil from Randfontein site, Rc-1–3 triplicate bulk (control) soil from Randfontein site.

**Figure 5 microorganisms-09-01419-f005:**
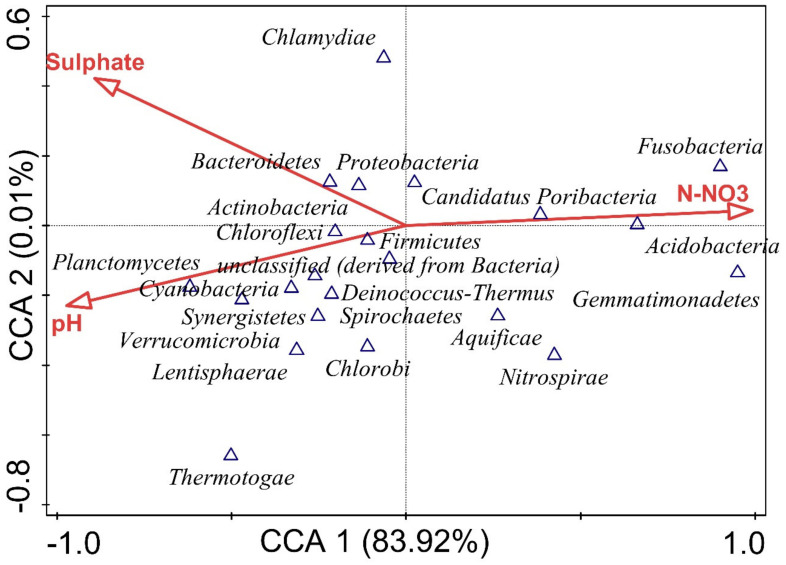
Canonical correspondence analysis (CCA) of bacterial phyla explained using forward selection of environmental variables that best predict the variation and influence on bacterial diversity.

**Table 1 microorganisms-09-01419-t001:** Physicochemical properties of maize rhizosphere and bulk soils.

Sample Locations	Physicochemical Values								
	pH (H_2_O)	Sulfate (mg/kg)	Total C (%)	P (mg/kg)	K (mg/kg)	N-NO_3_ (mg/kg)	N-NH_4_ (mg/kg)	Org C (%)	OM (%)
Ls	5.62 ± 0.09 ^a^	1.60 ± 1.68 ^a^	0.90 ± 0.05 ^a^	50.98 ± 1.77 ^a^	240.00 ± 2.94 ^a^	16.29 ± 2.25 ^a^	3.61 ± 0.29 ^ab^	0.61 ± 0.02 ^a^	3.40 ± 0.16 ^a^
Lc	5.87 ± 0.22 ^a^	0.44 ± 0.36 ^a^	0.90 ± 0.01 ^a^	65.86 ± 13.71 ^a^	243.00 ± 0.82 ^a^	16.24 ± 0.59 ^a^	2.42 ± 0.19 ^a^	0.60 ± 0.01 ^a^	3.25 ± 0.03 ^a^
Rs	6.76 ± 0.28 ^b^	2.56 ± 2.66 ^a^	1.34 ± 0.24 ^a^	257.14 ± 35.32 ^b^	167.00 ± 11.63 ^b^	8.52 ± 2.68 ^b^	2.91 ± 1.12 ^a^	1.09 ± 0.09 ^b^	3.43 ± 0.39 ^a^
Rc	6.73 ± 0.26 ^b^	2.32 ± 2.75 ^a^	0.85 ± 0.50 ^a^	206.54 ± 81.73 ^b^	148.50 ± 34.95 ^b^	7.38 ± 2.46 ^b^	4.75 ± 1.21 ^b^	0.87 ± 0.15 ^c^	2.95 ± 0.85 ^a^
*p*-values	˂0.000	0.623	0.187	0.001	˂0.000	0.001	0.044	˂0.000	0.609

Each value is expressed as mean ± standard deviation (n = 3). ^<a–c>^ indicates significant difference in values of samples according to Tukey’s HSD test (*p* ≤ 0.05). *p*-values given across the rows were used to compare chemical properties of the sampling sites. Ls—Lichtenburg rhizosphere sample, Lc—Lichtenburg bulk (control) sample, Rs—Randfontein rhizosphere sample, Rc—Randfontein bulk (control) sample. All statistical analyses, including mean values and analysis of variance (ANOVA) were done using GraphPad Prism (v5.0).

**Table 2 microorganisms-09-01419-t002:** Alpha diversity indices of bacterial communities across the sampling sites.

Level	Indices	Ls	Rs	Lc	Rc	*p*-Value
Bacterial phylum						
	Shannon_H	1.518	1.520	1.632	1.539	0.976
	Evenness_e ˆ H/S	0.207	0.208	0.233	0.212	
Class						
	Shannon_H	1.964	1.895	2.051	1.919	0.949
	Evenness_e ˆ H/S	0.274	0.256	0.299	0.262	
Order						
	Shannon_H	2.389	2.264	2.456	2.264	0.931
	Evenness_e ˆ H/S	0.363	0.321	0.389	0.321	
Family						
	Shannon_H	3.179	3.268	3.200	3.325	0.544
	Evenness_e ˆ H/S	0.687	0.751	0.701	0.794	
Genus						
	Shannon_H	3.109	3.155	3.064	3.188	0.860
	Evenness_e ˆ H/S	0.640	0.670	0.612	0.693	

*p-*Value was based on Kruskal-Wallis one-way analysis of variance. Ls—Lichtenburg rhizosphere sample, Lc—Lichtenburg bulk (control) sample, Rs—Randfontein rhizosphere sample, Rc—Randfontein bulk (control) sample.

**Table 3 microorganisms-09-01419-t003:** Influence of environmental variables on the bacterial composition of soil samples, as explained by the forward selection of environmental variables.

Physicochemical Parameter	Contribution (%)	Pseudo-F	*p*-Value
N-NO_3_	82.70	9.50	0.019
Sulfate	9.70	1.30	0.50
pH	7.60	˂0.1	1.00

**Table 4 microorganisms-09-01419-t004:** Comparison of important nutrient pathways and stress functional signatures obtained from the metagenome study of maize rhizosphere and bulk soils as annotated using SEED-subsystem database of MG-RAST.

SEED-Subsystem Level		Sampling Sites				
Level 2	Level 3	Ls	Rs	Lc	Rc	*p*-Value
**Nitrogen Metabolism**	**Total Hits**	**88,948**	**74,740**	**69,607**	**73,260**	
		**Relative Abundance (%)**				
Allantoin Utilization		3.43 ± 0.10 ^a^	3.11 ± 0.10 ^b^	3.32 ± 0.06 ^ab^	3.13 ± 0.10 ^b^	0.02
Ammonia Assimilation	Ammonia Assimilation	25.60 ± 0.37 ^a^	24.16 ± 0.27 ^b^	25.54 ± 0.27 ^a^	23.26 ± 0.41 ^b^	0.00
	Glutamate and Aspartate uptake in bacteria	4.88 ± 0.33 ^a^	3.99 ± 0.32 ^b^	3.87 ± 0.27 ^ab^	4.12 ± 0.49 ^ab^	0.05
	Glutamate dehydrogenase	4.21 ± 0.08 ^a^	3.86 ± 0.51 ^a^	4.15 ± 0.08 ^a^	3.83 ± 0.24 ^a^	0.46
Glutamine, Glutamate, Aspartate and Asparagine Biosynthesis		28.87 ± 0.14 ^a^	27.70 ± 0.30 ^a^	29.13 ± 0.10 ^a^	28.79 ± 1.57 ^a^	0.12
	Glutamine Synthetases	2.38 ± 0.03 ^ab^	2.28 ± 0.07 ^a^	2.49 ± 0.08 ^b^	2.50 ± 0.07 ^b^	0.02
	Poly-gama-glutamate Biosynthesis	2.65 ± 0.09 ^a^	2.19 ± 0.09 ^ab^	2.54 ± 0.12 ^a^	1.97 ± 0.35 ^b^	0.02
Cyanate Hydrolysis		0.71 ± 0.01 ^a^	0.76 ± 0.14 ^a^	0.70 ± 0.04 ^a^	0.76 ± 0.08 ^a^	0.82
Denitrification		1.95 ± 0.12 ^a^	2.72 ± 0.16 ^b^	5.40 ± 0.19 ^ab^	4.94 ± 0.35 ^ab^	0.01
Nitrate and Nitrite Ammonification		15.86 ± 0.30 ^a^	20.98 ± 0.23 ^b^	14.97 ± 0.41 ^a^	18.00 ± 3.44 ^ab^	0.01
Nitric Oxide Synthase		6.61 ± 0.13 ^a^	6.96 ± 0.43 ^a^	5.39 ± 0.02 ^b^	5.37 ± 0.30 ^b^	0.00
Nitrilase		0.08 ± 0.01 ^a^	0.05 ± 0.02 ^a^	0.05 ± 0.01 ^a^	0.05 ± 0.01 ^a^	0.11
Nitrogen Fixation		2.11 ± 0.07 ^ab^	2.35 ± 0.28 ^a^	1.52 ± 0.14 ^ab^	1.35 ± 0.52 ^b^	0.02
Nitrosative Stress		0.77 ± 0.07 ^a^	0.87 ± 0.14 ^a^	1.85 ± 0.04 ^b^	1.93 ± 0.13 ^b^	˂0.00
**Phosphorus Metabolism**	**Total Hits**	**55,847**	**44,263**	**44,281**	**43,346**	
		**Relative Abundance (%)**				
Alkylphosphonate Utilization		6.48 ± 0.13 ^a^	6.45 ± 0.53 ^a^	5.28 ± 0.23 ^b^	5.02 ± 0.29 ^b^	0.01
Alkylphosphonate (TC_3.A.1.9.1)		1.55 ± 0.07 ^a^	1.81 ± 0.28 ^a^	1.49 ± 0.09 ^a^	1.68 ± 0.01 ^a^	0.22
High Affinity Phosphate Transporter and Control of PHO Regulon		18.71 ± 0.12 ^a^	18.38 ± 0.15 ^a^	18.90 ± 0.35 ^a^	18.75 ± 0.50 ^a^	0.27
P uptake Cyanobacteria		8.07 ± 0.09 ^a^	8.06 ± 0.13 ^a^	6.92 ± 0.17 ^b^	6.99 ± 0.37 ^b^	0.00
Phosphate-Binding DING Proteins		0.39 ± 0.17 ^a^	0.19 ± 0.03 ^a^	0.14 ± 0.04 ^a^	0.17 ± 0.03 ^a^	0.10
Phosphate Metabolism	PhoR-PhoB two Component Regulatory	2.50 ± 0.11 ^a^	2.47 ± 0.01 ^a^	3.55 ± 0.10 ^a^	3.32 ± 0.08 ^a^	0.12
	Phosphate Metabolism	61.32 ± 0.23 ^a^	62.29 ± 1.12 ^a^	52.09 ± 0.38 ^b^	52.79 ± 1.04 ^b^	˂0.01
Phosphoenolpyruvate phosphomutase		0.74 ± 0.08 ^a^	0.58 ± 0.17 ^a^	0.58 ± 0.13 ^a^	0.55 ± 0.21 ^a^	0.48
Phosphonate Metabolism		1.25 ± 0.07 ^a^	0.77 ± 0.11 ^b^	1.05 ± 0.03 ^ab^	0.73 ± 0.37 ^ab^	0.04
**Iron Acquisition and Metabolism**	**Total Hits**	**26,796**	**21,718**	**18,688**	**20,450**	
		**Relative Abundance (%)**				
Siderophore	Siderophore_Achromobactin	0.17 ± 0.14 ^a^	0.06 ± 0.01 ^a^	0.04 ± 0.03 ^a^	0.08 ± 0.02 ^a^	0.35
	Siderophore_Aerobactin	0.09 ± 0.01 ^a^	0.18 ± 0.07 ^b^	0.06 ± 0.00 ^ab^	0.17 ± 0.03 ^ab^	0.05
	Siderophore_Desferrioxamine_E	0.38 ± 0.06 ^a^	0.76 ± 0.60 ^a^	0.27 ± 0.03 ^a^	0.53 ± 0.19 ^a^	0.48
	Siderophore_Enterobactin	0.28 ± 0.26 ^a^	0.16 ± 0.05 ^a^	0.11 ± 0.03 ^a^	0.19 ± 0.10 ^a^	0.68
	Siderophore_Pyoverdine	9.90 ± 0.73 ^a^	9.62 ± 1.05 ^a^	9.60 ± 0.10 ^a^	8.05 ± 2.08 ^a^	0.38
	Siderophore_Staphylobactin	0.06 ± 0.01 ^a^	0.02 ± 0.01 ^b^	0.07 ± 0.02 ^a^	0.01 ± 0.00 ^b^	0.00
	Siderophore_Yersiniabactin_Biosynthesis	1.27 ± 0.11 ^a^	1.53 ± 0.08 ^a^	1.41 ± 0.07 ^a^	1.19 ± 0.22 ^a^	0.08
	Siderophore_Assembly_Kit	5.96 ± 0.46 ^a^	8.21 ± 1.67 ^b^	5.89 ± 0.22 ^ab^	7.44 ± 0.25 ^ab^	0.05
	Siderophore_Pyochelin	0.22 ± 0.04 ^a^	0.28 ± 0.04 ^a^	0.19 ± 0.00 ^a^	0.25 ± 0.13 ^a^	0.47
	Vibrioferrin_Synthesis	0.17 ± 0.03 ^a^	0.12 ± 0.03 ^ab^	0.11 ± 0.02 ^ab^	0.05 ± 0.03 ^b^	0.02
	Bacillibactin_Siderophore	0.52 ± 0.05 ^a^	0.55 ± 0.03 ^a^	0.56 ± 0.03 ^a^	0.48 ± 0.08 ^a^	0.39
Transport of Iron		9.71 ± 0.54 ^ab^	11.40 ± 0.70 ^a c^	9.01 ± 0.73 ^b^	12.30 ± 0.69 ^c^	0.01
ABC Type Iron Transport System		3.26 ± 0.32 ^a^	1.88 ± 0.31 ^b^	3.19 ± 0.06 ^a^	1.86 ± 0.38 ^b^	0.00
ABC transporter Iron.B12.Siderophore.hemin		0.71 ± 0.03 ^a^	0.98 ± 0.15 ^ab^	0.61 ± 0.10 ^b^	0.75 ± 0.16 ^ab^	0.05
Campylobacter Iron Metabolism		12.16 ± 0.43 ^ab^	12.20 ± 0.70 ^a^	13.70 ± 0.29 ^b^	12.89 ± 0.06 ^ab^	0.04
Ferrous Iron Transporter EfeUOB Low pH Induced		4.77 ± 0.41 ^a^	5.30 ± 1.37 ^a^	4.45 ± 0.33 ^a^	5.70 ± 2.54 ^a^	0.78
Hemen Hemin uptake and Utilization Systems (Gram Negative)		8.93 ± 0.42 ^a^	7.67 ± 0.33 ^ab^	9.00 ± 0.45 ^b^	7.81 ± 0.91 ^ab^	0.05
Hemen Hemin uptake and Utilization Systems (Gram Positives)		3.52 ± 0.39 ^a^	3.55 ± 0.29 ^a^	4.11 ± 0.45 ^a^	3.76 ± 0.09 ^a^	0.31
Hemin Transport System		2.45 ± 0.21 ^a^	2.37 ± 0.34 ^a^	2.46 ± 0.06 ^a^	2.59 ± 0.18 ^a^	0.81
Iron III Dicitrate Transport System Fec		0.94 ± 0.11 ^a^	1.10 ± 0.08 ^a^	0.81 ± 0.10 ^a^	1.10 ± 0.24 ^a^	0.15
Iron Scavenging Cluster in *Thermus*		0.75 ± 0.07 ^a^	1.26 ± 0.27 ^b^	0.70 ± 0.00 ^a^	1.12 ± 0.12 ^ab^	0.02
Iron Acquisition in *Streptococcus*		1.64 ± 0.06 ^a^	2.51 ± 0.21 ^b^	1.74 ± 0.22 ^a^	2.68 ± 0.07 ^b^	0.00
Iron Acquisition in *Vibrio*		31.89 ± 1.48 ^a^	27.85 ± 2.06 ^ab^	31.66 ± 0.56 ^a^	28.64 ± 1.36 ^b^	0.05
**Stress Response**	**Total Hits**	**86,683**	**68,762**	**66,309**	**66,096**	
		**Relative Abundance (%)**				
SigmaB Stress Response Regulation		6.36 ± 0.23 ^a^	6.40 ± 0.27 ^a^	6.59 ± 0.35 ^a^	6.33 ± 0.17 ^a^	0.74
Osmotic Stress	Synthesis of Osmoregulation Periplasmic glucans	5.22 ± 0.13 ^a^	5.79 ± 1.30 ^a^	5.62 ± 0.07 ^a^	6.46 ± 1.88 ^a^	0.37
	Choline and Betaine Uptake and Biosynthesis	8.97 ± 0.46 ^a^	11.03 ± 0.62 ^b^	8.42 ± 0.61 ^a^	9.85 ± 0.96 ^ab^	0.02
	Ectoine Synthesis and Regulation	1.23 ± 0.05 ^a^	0.33 ± 0.11 ^b^	0.22 ± 0.07 ^b^	0.26 ± 0.07 ^b^	˂0.00
Osmoprotection ABC Transporter YehZYXW of Enterobacteriales		0.08 ± 0.04 ^a^	0.05 ± 0.01 ^a^	0.07 ± 0.00 ^a^	0.04 ± 0.01 ^a^	0.33
	Osmoregulation	1.70 ± 0.03 ^a^	2.02 ± 0.14 ^ab^	1.77 ± 0.01 ^ab^	2.08 ± 0.21 ^b^	0.03
Phage Shock Protein Operon		1.54 ± 0.01 ^a^	1.18 ± 0.22 ^b^	1.46 ± 0.06 ^ab^	1.22 ± 0.08 ^ab^	0.05
Oxidative Stress	CoA-disulfide Reductase (EC_1.8.1.14) cluster	1.28 ± 0.01 ^a^	1.33 ± 0.03 ^a^	0.26 ± 0.02 ^b^	0.30 ± 0.02 ^b^	0.04
	Rubrerythrin	2.18 ± 0.03 ^a^	2.05 ± 0.02 ^a^	2.23 ± 0.03 ^a^	2.06 ± 0.28 ^a^	0.34
	Glutaredoxins	0.38 ± 0.02 ^a^	0.40 ± 0.03 ^a^	0.37 ± 0.00 ^a^	0.37 ± 0.02 ^a^	0.44
	Glutathione: Biosynthesis and Gamma Glutamyl Cycle	9.45 ± 0.32 ^a^	8.26 ± 0.35 ^ab^	9.82 ± 0.04 ^ab^	8.02 ± 1.31 ^ab^	0.04
	Glutathione: non-redox Reaction	4.76 ± 0.22 ^a^	4.30 ± 0.16 ^ab^	4.24 ± 0.24 ^ab^	4.03 ± 0.18 ^b^	0.03
	Glutathione: redox Cycle	1.84 ± 0.19 ^a^	1.84 ± 0.08 ^a^	1.79 ± 0.07 ^a^	1.81 ± 0.13 ^a^	0.97
	Glutathione Analogs Mycothiol	3.95 ± 0.12 ^a^	4.60 ± 0.51 ^a^	3.78 ± 0.31 ^a^	5.43 ± 2.33 ^a^	0.39
	NADPH Quinone Oxidoreductase_2	0.38 ± 0.01 ^a^	0.31 ± 0.08 ^a^	0.36 ± 0.02 ^a^	0.38 ± 0.11 ^a^	0.57
	Protection from Reactive Oxygen Species	4.12 ± 0.10 ^a^	4.44 ± 0.31 ^a^	4.06 ± 0.11 ^a^	4.70 ± 0.57 ^a^	0.20
	Regulation of Oxidative Stress Response	10.21 ± 0.33 ^a^	9.75 ± 0.42 ^a^	10.25 ± 0.49 ^a^	9.31 ± 0.46 ^a^	0.15
Acid Stress	Acid Resistance Mechanisms	2.12 ± 0.05 ^a^	2.58 ± 0.30 ^a^	2.41 ± 0.19 ^a^	2.53 ± 0.41 ^a^	0.23
Glutamate Transporter involved in Acid tolerance in *Streptococcus*		0.07 ± 0.02 ^a^	0.13 ± 0.01 ^a^	0.07 ± 0.01 ^a^	0.17 ± 0.08 ^a^	0.06
Detoxification (Toxin Stress)	Uptake of Selenate and Selenite	3.85 ± 0.24 ^a^	3.54 ± 0.21 ^a^	3.32 ± 0.41 ^a^	3.73 ± 0.34 ^a^	0.34
Glutathione-dependent Pathway of Formaldehyde Detoxification		1.91 ± 0.09 ^a^	1.72 ± 0.13 ^ab^	1.80 ± 0.34 ^ab^	1.57 ± 0.63 ^b^	0.03
	Tellurite Resistance Chromosomal Determinants	0.03 ± 0.04 ^a^	0.01 ± 0.07 ^b^	0.01 ± 0.18 ^b^	0.02 ± 0.01 ^ab^	0.02
Cold Shock	Cold Shock *CspA* Family of Proteins	3.11 ± 0.00 ^a^	2.61 ± 0.00 ^b^	2.12 ± 0.01 ^a c^	2.70 ± 0.01 ^b c^	0.01
Heat Shock	Heat Shock *dnaK* Gene Cluster Extended	20.49 ± 0.07 ^a^	20.37 ± 0.11 ^a^	21.40 ± 0.14 ^a^	20.74 ± 0.19 ^a^	0.08

Each value is expressed as mean ± standard deviation (n = 3). ^<a–c>^ indicates significant difference in values of samples according to Tukey’s HSD test (*p* ≤ 0.05). Ls—Lichtenburg rhizosphere sample, Lc—Lichtenburg bulk (control) sample, Rs—Randfontein rhizosphere sample, Rc–Randfontein bulk (control) sample.

## Data Availability

The raw sequences are publicly available on NCBI SRA database with project identification number PRJNA645385 and PRJNA645371for Ls and Rs sites, respectively.

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
