# Peer review of "Metagenomic Insight into the Community Structure of Maize-Rhizosphere Bacteria as Predicted by Different Environmental Factors and Their Functioning within Plant Proximity"

_microorganisms, 2021, doi:10.3390/microorganisms9071419_

Round 1

Reviewer 1 Report

The discussion section should be improved according to commnets.

Author Response

Response to Reviewer 1 Comments

Point 1: This sentence should be re-written because is not clear (Line 47-48).

Response 1: The statement has been rephrased as ‘Thus, microbe inhabiting the plant tissues are regulated by various mechanisms controlled by the host plant.

Point 2: This paragraph is the same in results, it is not a discussion (439-444)

Response 2: The statement has been rephrased to capture only important part of the result.

Point 3: This could be an interesting discussion, but unfortunatley it si not complete and not referring to other genera. Even in the paper are interesting functionala analysis, there is not a integrative discussion of the results. 

Response 3: Only the important (significant) genera were highlighted to integrate the result with discussion as you have suggested and compared to other studies. Discussing other genera common to all sites will surely invalidate the essence of the write-up (comparing samples) and likewise lead to repetition. Also, a research by Garrido-Sanz et al (2020) was used to buttress the statement immediately after the paraphrasing.

Point 4: I would rather discuss the potential of rhisoremediation in more common pollutants as excess fertilizers, heavy metals

Response 4: Discussion related to ‘heavy metal removal’ has been added (Line 463-464).

Point 5: as it is mentioned here, it would be interesting to discuss the results according to this particular pollution source.

Response 5: Reference and comment related to heavy metal inserted (Line 463-464)

Point 6: This is also described in results

Response 6: Statement now summarized.

Point 7: as the most important fingings of this paper si the particular structure of microorgansims communities in rhizosphere and bulk samples, it would be interesting to discuss the results in conection with their functional profile. Also, it would be interesting to highlight a correlation between community structure, functional profile and agronomical value of the soil if possible.

Response 7: Only transcriptomics could be used to correlate the above highlighted parameters. With the result deduced from this metadata, a reasonable inference was drawn (581-584). Using metagenomics, only summary of activities related to each sample could be identified. For specificity, metatranscriptomics will be the most appropriate technique to unveil microorganisms responsible for specific functions.

Point 8: Most of the results are presented again in discussion!

Response 8: The statement has been deleted

Point 9: it would be interesting to add references about bacteria as well!

Response 9: Reference related to bacteria has been added. ‘This research echoes the research of Molefe et al. [63] on the bacterial community structure and functions at the rhizosphere of maize plant’

Reviewer 2 Report

Comments:

The article is interesting with high utilitarian and substantive value. The authors' implementation of molecular biology methods and techniques as well as bioinformatic processing methods, combined with the interesting and insightful discussion, resulted in a really good scientific article.

I have two minor comments:

- look again at the notation of units - make it uniform and correct throughout the text;

- expand the last sentence in conclusions: „Understanding the mechanisms involved in rhizosphere-bacterial community selection, especially the importance of rhizodeposits in modification of soil microbiome is needed to explore the benefits of agroecosystem”.

Author Response

Response to Reviewer 2 Comments

Point 1: look again at the notation of units - make it uniform and correct throughout the text.

Response 1: The units are now uniform

Point 2: Expand the last sentence in conclusions: „Understanding the mechanisms involved in rhizosphere-bacterial community selection, especially the importance of rhizodeposits in modification of soil microbiome is needed to explore the benefits of agroecosystem”.

Response 2: The statement has been explained